# Experimental–Computational Investigation of the Elastic Modulus of Mortar under Sulfate Attack

**DOI:** 10.3390/ma16186167

**Published:** 2023-09-12

**Authors:** Zhongzheng Guan, Haochang Zhang, Yan Gao, Pengfei Song, Yong Li, Lipeng Wu, Yichao Wang

**Affiliations:** Key Laboratory of Roads and Railway Engineering Safety Control, Ministry of Education, Shijiazhuang Tiedao University, Shijiazhuang 050013, China; guanzhongzheng@stdu.edu.cn (Z.G.); gaoyan2023@foxmail.com (Y.G.);

**Keywords:** sulfate attack, mortar, elastic modulus, nanoindentation technique, homogenization methods

## Abstract

External sulfate attack is an important factor causing a decrease in the mechanical properties of cement-based materials. In this paper, a computational prediction model of elastic modulus, considering the characteristics of sulfate corrosion from outside to inside and the influence of the interface transition zone (ITZ), was established to predict the elastic modulus of mortar under the external sulfate attack. Firstly, the backscattered electron (BSE) images of mortar and the algorithm of image threshold segmenting were used to determine a reasonable thickness of corroded ITZ. Secondly, the nanoindentation test was adopted to acquire the microscopic elastic parameters of phases (sand, cement, and ITZ) in corroded mortar. Moreover, the mortar mix proportion and Lu and Torquato’s model were adopted to calculate the volume fractions of phases. Finally, a computational prediction model of elastic modulus of mortar under sulfate attack was proposed with homogenization methods. The results indicate that the thickness of corroded ITZ is 20 mm, and the error values of elastic modulus between the theoretical prediction results and the experimental results are within 8%, indicating that the macroscopic elastic modulus of corroded mortar can be precisely predicted by the computational prediction model of elastic modulus.

## 1. Introduction

Elastic modulus has an important influence on the stress distribution and deformation of concrete structures, and, as such, it is a key parameter in the design of such structures [1]. Sulfate corrosion in environments, such as oceans, salt lakes, and groundwater, is an important factor causing durability issues and structural failure of cement materials [2]. The durability of cement-based material under sulfate attack has attracted widespread attention and research from scholars [3]. Mortar is between cement paste and concrete, and the elastic modulus of mortar is an important parameter connecting the elastic modulus of concrete and cement. Therefore, the study of the mortar’s elastic modulus under sulfate attack has a great significance.

As is known, the macroscopic mechanical behavior of material is largely dependent on the compositions, contents, and mechanical properties of phases at the nano/micro-scale [4]. The upscaling technique was accurately adopted to calculate the elastic parameters of material between the microscopic and macroscopic properties over the last several decades [5,6,7,8,9]. A multi-scale mechanical model was established with the Mori–Tanaka (M-T) method and self-consistent (SC) method to precisely predict the intrinsic elastic properties of concrete [10]. According to the results of the nanoindentation test, Li et al. divided the brine-eroded cement paste into two scales: C-S-H and cement paste [11]. The M-T method and SC method were used to predict the elastic modulus of C-S-H gel and cement paste after brine corrosion. Based on the cement hydration model, Huang et al. and Liang et al. used the multi-scale homogenization method to predict the elastic modulus of concrete from the C-S-H scale [1,7]. However, there are few studies on the elastic modulus of sulfate corrosive mortar with multi-scale homogenization methods.

The interface transition zone (ITZ) is between the matrix and the inclusions. The ITZ is the weakest phase in concrete material, with the characteristics of low elastic modulus, low strength, high porosity, and high permeability [12,13,14]. The macroscopic mechanical behavior of cement-based materials is largely dependent on the component content and mechanical properties of phases. Considering the influence of ITZ, Liang et al. divided the concrete into four scales and predicted the concrete’s elastic modulus with the homogenization methods [1]. The ITZ’s effective thickness is a key parameter for calculating the volume fraction of phases, and the macroscopic mechanical performances of concrete [15]. However, the thickness of the ITZ was not considered in Liang’s model. In order to evaluate the effective thickness of the ITZ, various microstructure of cement-based materials have been observed and researched, and the thickness of the ITZ from 15 μm to 50 μm is usually accepted [15]. Lutz et al. considered that the ITZ around the concrete aggregate was non-uniform, and it increased slowly with the increase in the distance to the aggregate’s center [16]. It is considered that the porosity of the ITZ is closer to the porosity of the matrix the farther away from the aggregate, and then the elastic modulus of concrete was studied by assuming the elastic modulus of the ITZ. Mondal et al. studied the elastic properties of the ITZ with the nanoindentation technology and found that the change in the elastic modulus of the indentation point in ITZ is regular, that is, the elastic modulus increases gradually as the indentation point approaches the center of the fine aggregate [17,18]. The average value of elastic modulus of ITZ was 18 GPa, about 85% of the value of the matrix. Chen et al. adopted the nearest surface distribution function of Lu and Torquato’s model to give a quantitative calculation formula for calculating the volume fraction of the ITZ between aggregate and matrix in cement-based composites, then used the random point sampling method to calculate the volume fraction of ITZ for verifying the effectiveness of the quantitative calculation formula [19,20]. Although the existing studies have calculated the influence of the ITZ on the overall performance of cement-based materials under different thicknesses with the help of formulas, they lack the experimental verification of the elastic performance of the ITZ and do not involve the influence law of the mortar ITZ on the overall elastic energy of mortar under a sulfate corrosion environment.

In this paper, a reasonable thickness of ITZ in corroded mortar was obtained using image processing software with the SEM-BSE image of mortar. Then, the elastic parameters of phases (sand, cement, and ITZ) in mortar were determined based on the nanoindentation test and theoretical calculation. In addition, the volume fraction of each phase in the mortar was determined by calculating the mortar mix ratio and applying Lu and Torquato’s model. Finally, the homogenization method was used to establish the elastic modulus prediction model of mortar under sulfate attack. The theoretical calculation results were compared with the experimental results to verify the effectiveness of the elastic modulus prediction model. The research route is shown in Figure 1.

## 2. Materials and Methods

### 2.1. Materials

P.I 52.5 Portland cement was selected in this study. The mineral composition of the cement is shown in Table 1. The natural river sand was selected as the fine aggregate, with a minimum size of 0.035 mm, a maximum size of 4.75 mm, a fineness modulus of 2.45, and a mud content (by mass) < 1.5%.

Sodium sulfate (Na_2_SO_4_) was selected to prepare the sulfate solution, and the molar concentration of sulfate ion in sulfate solution was 352 mol/m^3^ (mass concentration was 5%).

### 2.2. Mix Design and Samples Preparation

In this study, the mortar mixing ratio design is shown in Table 2.

The mixed mortar was poured into 20 mm × 20 mm × 40 mm molds, cured in a laboratory with a constant temperature (20 ± 5 °C) and humidity (65%) for 24 h, and then demolded. Subsequently, the mortar samples were cured under standard curing conditions (temperature of 20 °C and relative humidity of 95%) for 28 d. Finally, mortar samples were soaked in Na_2_SO_4_ solution at 20 ± 2 °C. In order to ensure a constant concentration of sulfate ions, the Na_2_SO_4_ solution was replaced every 3 d.

The uniaxial compression test, the nanoindentation test, and the SEM-BSE test were conducted, and mortar samples were selected with a processing period of 88 d (sulfate corrosion time of 60 d). For the uniaxial compression test, each mix proportion selected five mortar specimens to obtain the constitutive relation curves. For the nanoindentation test, specimens were selected from mortar specimens with dimensions of 5 mm × 5 mm × 10 mm. Owing to the high requirement of flatness for the nanoindentation test, nanoindentation specimens were pretreated [21,22]. Firstly, in order to prevent the further hydration of the specimen, the specimen was placed in an alcohol phenolphthalein solution, and then the specimen was taken out and placed in the mold, which was carried out with epoxy resin. Then, in the coarse polishing process, the specimens were polished on the grinding and polishing machine using abrasive paper of 180 meshes, 240 meshes, 400 meshes, 600 meshes, 800 meshes, and 1200 meshes, respectively. Then, the specimens were polished using canvas and silk cloth with a suspension of 0.25 μm diamond. It should be noted that the upper and lower planes of specimens must be parallel during the grinding process. Furthermore, atomic force microscopy (AFM) was used to measure the surface roughness of the polished specimens. The polishing process ended when the surface roughness was less than 100 nm. Finally, the specimens were washed in an ultrasonic bath with anhydrous ethanol to remove the suspensions and debris on the specimen surface, and then the specimens were dried naturally. The specimens for the nanoindentation test with a thickness of 4 mm are shown in Figure 2. The SEM-BSE test specimens were selected from the specimens after the nanoindentation test with additional treatment. After the nanoindentation test, specimens were sprayed with gold for 120 s. Then, a gold sprayed film with a thickness of 10 nm was formed on the surface of the specimen to achieve a better observation effect.

### 2.3. Experimental Methods

#### 2.3.1. Uniaxial Compression Test

The uniaxial compression experiments were conducted by a servo-controlled setup (Zwick/Roell Z050) with a loading speed of 0.5 mm/min. Two linear variable differential transformers (LVDT1 and LVDT2) were symmetrically arranged 10 mm away from two sides of the specimen to measure the deformation under loading. Figure 3 shows the uniaxial compression test of the mortar specimen. In order to increase the accuracy of experimental results, the maximum and minimum elastic modulus obtained from uniaxial compression experiments were eliminated in each group with the other three left. Then, the three remaining test dates of each group were used to calculate the macroscopic static compressive elastic modulus through the method of taking the average value.

#### 2.3.2. Nanoindentation Test

Instrumented nanoindentation (G200), produced by Agilent Company, Santa Clara, CA, USA, was adopted to acquire the microstructural mechanical properties of corroded mortar specimens. The elastic modulus and hardness of the specimen varying with the nanoindentation depth were obtained by a continuous stiffness method (CSM). The schematic of the nanoindentation test for the microscopic elastic modulus of the corroded mortar is shown in Figure 4.

During the loading process, elastic deformation firstly appeared in the specimen. As for the plastic deformation of the specimen, the loading curve was non-linear with the increase in loading. During unloading, only the elastic properties of the indentation point were restored. Figure 5 shows the load(P)–displacement(H) curve during the typical loading and unloading of the indenter. The curve was used to analysis the hardness and elastic modulus of indentation points. The microscopic elastic modulus E can be calculated by the Oliver–Pharr principle [23], as shown in the following Equation (1):(1)$$E=\frac{1-\nu^{2}}{\frac{2\beta }{S}\sqrt{\frac{A}{\pi }}-\frac{1-\nu_{i}^{2}}{E_{i}}}$$
where *E* is the elastic modulus of tested material, ν is the Poisson’s ratio of tested material, β represents a correction factor (due to the Berkovich indenter, β = 1.034), *S* is the contact stiffness ($S={\left(dP/dh\right)}_{h=h_{\max}}$, *P* is the maximum loading), *A* is the corresponding contact area at the maximum load, $\nu_{i}$, and $E_{i}$ denotes the parameters of indenter ($\nu_{i}$ = 0.07, $E_{i}$ = 1141 GPa).

The nanoindentation test was carried out on ITZ, cement, and sand of the mortar specimens. An 8 × 8 lattices of 64 points on the corroded mortar specimen was measured to obtain the microscopic elastic modulus.

In order to obtain a relatively stable elastic modulus value, the indentation depth needs to reach the micrometer scale [1]. Moreover, it can be more clearly observed by the scanning electron microscope (SEM) with the increase in the indentation depth. Then, the surface area of the indentation point will increase and the possibility of interaction between two adjacent points will increase. The ITZ’s thickness in concrete was 15 µm to 50 µm, and the ITZ’s thickness in mortar was smaller than that in concrete [24,25,26]. For the sake of the accurate elastic modulus of ITZ, the closer the distance between every two adjacent nanoindentation points, the better the test result, at least theoretically. However, there should be a sufficient distance between two adjacent indentation points to avoid mutual influence. Therefore, the depth of nanoindentation points and the spacing of nanoindentation points were set as 2000 nm and 15 µm, respectively.

#### 2.3.3. SEM-BSE Test

The micromorphology of nanoindentation points and corroded mortar was observed by backscattered electron (SEM-BSE) images. The SEM-BSE images were obtained by Quanta 250FEG, produced by FEI Co., Hillsboro, OR, USA. The SEM-BSE images were acquired with a resolution of 3.5 nm under a high vacuum mode, and the distance between the sample and scanning electron microscope lens was 10 mm. Figure 6 shows the micromorphology of a lattice map of nanoindentation points and a single nanoindentation point of cement mortar (M1) observed by the SEM-BSE test. The region in the red box in Figure 6a is the lattice map of nanoindentation points, and the region in the blue box in Figure 6b is a single nanoindentation point.

## 3. Experimental Results and Discussion

### 3.1. Macroscopic Elastic Modulus of Corroded Mortar

The stress–strain curve of the eroded mortar sample was obtained via a uniaxial compression test, so that the macroscopic elastic modulus (E) of the sample could be obtained. The macroscopic elastic moduli of corroded mortars with types of M1 and M2 were 29.88 GPa and 33.33 GPa, respectively.

### 3.2. Thickness of ITZ in Corroded Mortar

The BSE images of corroded mortar were processed by the algorithm of image threshold segmenting. Each of cubic volume elements (voxels) in the image was assigned a unique grayscale value, then the data were organized as a collection of cubic volume elements (voxels).

Next, the data of distinct material phases were segmented into several distinct categories [7]. The material phases, such as the pores and matrix in the mortar, were distinguished and quantified by the threshold intensity segmentation method [27].

Figure 7 shows the SEM-BSE image processing process of ITZ in mortar (M1) at 88 d. Figure 7a shows the SEM-BSE image of corroded mortar. The yellow line in the figure is the edge of sand. After selecting the region of 60 µm at the edge of sand and dividing it into 12 regions successively, the thickness of each region was 5 µm. Then, median filtering and image binarization technology were used to process the BSE image to distinguish the pores and matrix [28]. According to the previous research results, the gray segmentation threshold between matrix and pores was 72.1 [29]. Figure 7b shows the binarization results of BSE images. In each region, blue parts are micropore structures, and the left part is the matrix. According to the pixel number of the matrixes and micropore structures in each region, the porosity of regions at different distances from the sand edge can be calculated by Equation (2), as follows:(2)$$\phi =\frac{n_{p}}{n}\times 100\%$$
where ϕ is the porosity of the image processing region, $n_{p}$ is the pixel number of micropore structures, and *n* is the total number of pixels in the processing region.

Firstly, the porosity of each region was calculated, and then 10 regions with the same distance to the edge of the sand are selected, and the average porosity of these 10 regions was calculated as the porosity of the region. According to Equation (2), the variation in porosity with the increase in the distance to the edge of the fine aggregate can be obtained, as shown in Figure 8. The porosity first decreases rapidly and then gradually tends towards a stable value with the increase in the distance from the sand edge. According to the previous study of Lutz, the porosity of cement in mortar is a stable value [16]. The porosity of ITZ in mortar gradually decreases with the increase in the distance from the surface of the sand, and finally tends to the value of the cement porosity [16]. Therefore, the ITZ thickness of corroded mortar specimens is about 20 µm.

### 3.3. Nanoindentation Results of Corroded Mortar

A lattice of 8 × 8 nanoindentation points, including the sand, ITZ, and paste, was carried out around the fine aggregate in corroded mortar. Figure 9 shows the SEM-BSE image and elastic modulus results of nanoindentation points of mortar specimen (M1) at 88 d. The microscopic elastic modulus of each point is marked on Figure 9b, corresponding to the locations. It is seen that in the gray regions (sands) in Figure 9b, the minimum value of the micro-elastic modulus is about 55 GPa, the maximum value is about 70 Gpa, and the average value is 63.1 Gpa. According to the analysis in Section 3.2, the area within 20 µm at the edge of sands is ITZ. It is seen that the microscopic elastic modulus of nanoindentation points in yellow regions (ITZ) range from 11 Gpa to 28 Gpa with an average value of 14.8 Gpa. The microscopic elastic modulus of cement paste ranges from 12 Gpa to 28 Gpa, with an average value of 21.9 Gpa.

Figure 10 shows the SEM-BSE image and the elastic modulus results of nanoindentation points of mortar specimen (M2) at 88 d. The gray regions in Figure 10b represent sands, and the elastic modulus of sand is close to that of sand in mortar (M1). The minimum value of the micro-elastic modulus of sand is about 55 Gpa, the maximum value is about 70 Gpa, and the average value is 63.4 Gpa. It can be found that the microscopic elastic modulus of nanoindentation points, within 20 µm at the edge of sands on yellow regions (ITZ) in Figure 10b, range from 13 Gpa to 28 Gpa, with an average value of 18.4 Gpa. The microscopic elastic modulus of cement paste ranges from 10 Gpa to 34 Gpa, and the average value is 23.0 Gpa.

## 4. Evaluation of the Elastic Modulus of Corroded Mortar by Homogenization Method

As known, the macro-elastic modulus of mortar can be measured by the static compressive test, and the macro-mechanical properties are mainly dependent on micromechanical properties and micro-components. The homogenization methods can be used to calculate elastic modulus of mortar from multi-scale for studying the influence of sulfate corrosion on the mortar.

### 4.1. Homogenization Method

The corroded mortar can be divided into two levels according to the multi-scale method, as shown in Figure 11.

On level I, the cement paste matrix, with a characteristic length range from 10^−6^ m to 10^−4^ m, is composed of unhydrated cement particles (C3S, C2S, C3A, and C4AF), hydrated products (LD C-S-H, HD C-S-H, CH, C4AH13, and C3 (A,F)H6), sulfate corrosion products (Aft), and pores, etc. The elastic modulus of cement paste can be obtained by the micromechanical model and homogenization method from the micro-elastic modulus of each phase in cement paste. As is known, when the reference medium in composites was hard to determine, the self-consistent (SC) method can be selected as an effective homogenization method [30,31,32]. In the SC method, the homogenized medium is considered as the reference medium. In this study, cement paste was regarded as a material containing 10 different phases, and the SC method was adopted as the homogenization method. The bulk modulus and shear modulus of the cement paste matrix can be calculated according to Equations (3) and (4), as follows [31,32]:(3)$$k_{SC}^{\mathrm{hom}}=k_{0}+\sum_{r=1}^{N}\frac{c_{r}(k_{r}-k_{0})(3k_{SC}^{\mathrm{hom}}+4\mu_{SC}^{\mathrm{hom}})}{3k_{r}+4\mu_{SC}^{\mathrm{hom}}}$$
(4)$$\mu_{SC}^{\mathrm{hom}}=\mu_{0}+\sum_{r=1}^{N}\frac{5c_{r}\mu_{SC}^{\mathrm{hom}}(\mu_{r}-\mu_{0})(3k_{SC}^{\mathrm{hom}}+4\mu_{SC}^{\mathrm{hom}})}{3k_{SC}^{\mathrm{hom}}(3\mu_{SC}^{\mathrm{hom}}+2\mu_{r})+4\mu_{SC}^{\mathrm{hom}}(2\mu_{SC}^{\mathrm{hom}}+3\mu_{r})}$$
where $k_{SC}^{\mathrm{hom}}$ and $\mu_{SC}^{\mathrm{hom}}$ represent the homogenized bulk modulus and shear modulus, respectively.

On level II, two kinds of multi-level homogenization methods were considered, as shown in Figure 12, to predict the macro-effective elastic parameters of mortar under sulfate attack. Mortar was treated as a composite material consisting of sand, ITZ, and cement paste, as shown in Figure 12a. For the first step, the near-field ITZ and far-field cement paste were considered equivalent by the SC method to form an equivalent matrix, as shown in Figure 12b. For the second step, the mortar was composed of the equivalent matrix and sand, and the macro-effective elastic modulus of mortar can be estimated by the two-phase sphere model, as rendered in Figure 12c.

Using the elastic method and Eshelby equivalent medium method [32,33], Christensen [34,35] proposed the following formula for solving the effective bulk modulus in the two-phase sphere model:(5)$${\bar{k}}_{s}=k_{E}+\frac{c_{s}(k_{s}-k_{E})(3k_{E}+4\mu_{E})}{3k_{E}+4\mu_{E}+3(1-c_{s})(k_{s}-k_{E})}$$
where ${\bar{k}}_{s}$ is the bulk modulus of equivalent particles, $k_{s}$ is the bulk modulus of sand, $k_{E}$ is the bulk modulus of equivalent matrix, $\mu_{E}$ is the shear modulus of equivalent matrix, and $c_{s}$ is the volume fraction of sand, which can be obtained from the following Equation (6):(6)$$c_{s}={\left(\frac{a}{b}\right)}^{3}=\frac{f_{s}}{f_{s}+f_{E}}$$
where $f_{s}$ and $f_{E}$ are the volume fractions of sand and equivalent matrix, respectively.

The effective shear modulus of equivalent particles can be obtained from the following Equations (7)–(14) [36]:(7)$$A{\left(\frac{{\bar{\mu }}_{s}}{\mu_{E}}\right)}^{2}+2B(\frac{{\bar{\mu }}_{s}}{\mu_{E}})+C=0$$
where
(8)$$\begin{array}{l} A=8(\frac{\mu_{s}}{\mu_{E}}-1)(4-5\nu_{E})\eta_{1}c^{10/3}-2{\left[63(\frac{\mu_{s}}{\mu_{E}}-1)\eta_{2}+2\eta_{1}\eta_{3}\right]}^{7/3}+\\ 252(\frac{\mu_{s}}{\mu_{E}}-1)\eta_{2}c^{5/3}-50(\frac{\mu_{s}}{\mu_{E}}-1)(7-12\nu_{E}+8\nu_{E}^{2})\eta_{2}c+4(7-10\nu_{E})\eta_{2}\eta_{3} \end{array}$$
(9)$$\begin{array}{l} B=-2(\frac{\mu_{s}}{\mu_{E}}-1)(4-5\nu_{E})\eta_{1}c^{10/3}-2{\left[63(\frac{\mu_{s}}{\mu_{E}}-1)\eta_{2}+2\eta_{1}\eta_{3}\right]}^{7/3}-\\ 252(\frac{\mu_{s}}{\mu_{E}}-1)\eta_{2}c^{5/3}+75(\frac{\mu_{s}}{\mu_{E}}-1)(3-\nu_{E})\nu_{E}\eta_{2}c+3(15\nu_{E}-7)\eta_{2}\eta_{3} \end{array}$$
(10)$$\begin{array}{l} C=4(\frac{\mu_{s}}{\mu_{E}}-1)(5\nu_{E}-7)\eta_{1}c^{10/3}-2{\left[63(\frac{\mu_{s}}{\mu_{E}}-1)\eta_{2}+2\eta_{1}\eta_{3}\right]}^{7/3}+\\ 252(\frac{\mu_{s}}{\mu_{E}}-1)\eta_{2}c^{5/3}+25(\frac{\mu_{s}}{\mu_{E}}-1)(\nu_{E}^{2}-7)\eta_{2}c-7(7+5\nu_{E})\eta_{2}\eta_{3} \end{array}$$
(11)$$\eta_{1}=(\frac{\mu_{s}}{\mu_{E}}-1)(49-50\nu_{s}\nu_{E})+35(\frac{\mu_{s}}{\mu_{E}})(\nu_{s}-2\nu_{E})+35(2\nu_{s}-\nu_{E})$$
(12)$$\eta_{2}=5(\frac{\mu_{s}}{\mu_{E}}-8)+7(\frac{\mu_{s}}{\mu_{E}}+4)$$
(13)$$\eta_{3}=(\frac{\mu_{s}}{\mu_{E}})(8-10\nu_{E})+(7-5\nu_{E})$$
(14)$${\bar{\mu }}_{s}=\frac{-B\pm \sqrt{B^{2}-AC}}{A}\mu_{E}$$
where *A*, *B*, *C*, $\eta_{1}$, $\eta_{2}$, and $\eta_{3}$ are the coefficients of the shear modulus solving process, ${\bar{\mu }}_{s}$ is the shear modulus of the equivalent particle, and $\mu_{E}$ is the shear modulus of the equivalent matrix.

According to Equations (5)–(14), the bulk modulus $k^{\mathrm{hom}}$ and shear modulus $\mu^{\mathrm{hom}}$ of mortar can be obtained from the micro-elastic parameters of sand and equivalent matrix. Then, the macro-elastic modulus of mortar obtained by homogenization methods can be calculated by the following Equation (15):(15)$$E^{\mathrm{hom}}=\frac{9k^{\mathrm{hom}}\mu^{\mathrm{hom}}}{3k^{\mathrm{hom}}+\mu^{\mathrm{hom}}}$$
where $E^{\mathrm{hom}}$ represents the elastic modulus of mortar, and $k^{\mathrm{hom}}$ and $\mu^{\mathrm{hom}}$ are the bulk modulus and shear modulus of mortar, respectively.

### 4.2. Mechanical Parameters of Phases

Under the corrosion of an external sulfate environment, sulfate ions enter the mortar through micropore structures and react with cement hydration products to produce expansive products, leading to the change in the mechanical properties of mortar [37,38,39,40].

It is generally believed that sulfate ions will not react with sand in mortar [41]. Therefore, it is assumed that the mechanical properties of sand do not change. It can be seen from the nanoindentation test results in Section 3.3 that the elastic modulus of sand in corroded mortar with different types has little difference. Then, the average elastic modulus of sand with the value of 63.25 Gpa was selected for calculating the macro-effective elastic parameters of the mortar. Furthermore, Poisson’s ratio of sand was 0.14 [41]. The shear modulus and bulk modulus of the sand can be calculated by Equations (16) and (17), and the values were 27.74 Gpa and 29.28 Gpa, respectively. Equations (16) and (17) are as follows:(16)$$\mu =\frac{E}{2(1+\nu )}$$
(17)$$k=\frac{E}{3(1-2\nu )}$$

In a sulfate environment, sulfate ions entering into mortar specimens and cause the corrosion of cement-based materials as part of a gradually decreasing process from outside to inside. Therefore, the corrosion degree of mortar was uneven inside and outside. Considering the influence of temporal and spatial distribution for the process of sulfate attack on mortar, the author of this paper coupled the effects of cement hydration and sulfate corrosion and established an X-CT-hydration–deterioration model [42]. On level I, the corroded cement paste matrix was regarded as a composite material composed of 10 phases on the micro-scale. Based on the X-CT-hydration–deterioration model, the compositions and volume fractions of microphases in cement-based material under sulfate attack were quantitatively predicted [42]. The elastic parameters of microphases in cement were obtained through a large number of nanoindentation experiments [29,43,44,45,46,47]. Then, through the SC method, the elastic parameters of cement paste were calculated with volume fractions and the elastic parameters of microphases [29]. The theoretical calculation results of elastic parameters of cement paste at 88 d are shown in Table 3 [29]. The theoretical calculation results were the output parameters of the elastic parameters of the cement paste on Level I, and also the input parameters of the elastic parameters of the cement paste on Level II.

According to the results of the nanoindentation tests in Section 3.3, the ratio of microscopic elastic modulus of ITZ to that of cement in the same corrosion layer of corroded mortar (M1) was about 0.7, and the value of that with type M2 was 0.8. Mondal et al. found that the elastic modulus ratio of ITZ and matrix was stable at a constant value [17,18]. Then, it was assumed that the microscopic elastic modulus ratio of ITZ and cement in corroded mortar (M1) remained at a constant value of 0.7, and the value of that with type M2 was 0.8. Therefore, according to the macro-elastic modulus of cement paste in Table 3, the macro-elastic modulus of ITZ in the corroded mortar (M1) was 15.2 Gpa, and the macro-elastic modulus of ITZ in the corroded mortar (M2) was 18.9 Gpa. Furthermore, Poisson’s ratio of ITZ was 0.24. Then, the bulk modulus and the shear modulus of ITZ in the corroded mortar (M1) were 9.74 Gpa and 6.13 Gpa, respectively. In the corrosion mortar (M2), the bulk modulus of ITZ is 12.12 Gpa, and the shear modulus is 7.62 Gpa.

### 4.3. Calculation of Volume Fractions of Microphases

According to the mix ratio of mortar, the volume fraction of sand in corroded mortar can be obtained by the following Equation (18):(18)$$f_{s}=\frac{V_{s}}{V_{w0}+V_{c0}+V_{s}}=\frac{m_{s}/\rho_{s}}{m_{w}/\rho_{w}+m_{c}/\rho_{c}+m_{s}/\rho_{s}}$$

Where $f_{s}$ is the volume fraction of sand in the mortar, $V_{s}$ is the volume of sand in the mortar, $V_{w0}$ and $V_{c0}$ are the volume of initial water and cement in the mortar, respectively, $m_{s}$, $m_{w}$, and $m_{c}$ are the mass content of sand, water, and cement, respectively, $\rho_{s}$, $\rho_{w}$ and $\rho_{c}$ the density of sand, water and cement, respectively.

The volume fractions of cement paste matrix (not including ITZ) and ITZ in the mortar were calculated by Lu and Torquato’s model, as shown in Figure 13 [48,49]. The model assumed the following. Firstly, fine aggregates were circular particles with different scales, randomly distributed in matrix. Secondly, the ITZ was a shell wrapping around the circular fine aggregate with a thickness of *t*, and the overlapping of the ITZ between adjacent aggregates was considered. Thirdly, the remaining part in the mortar was the cement paste matrix.

Based on the above assumptions, the volume fraction of the cement paste matrix was *e_v_*(*t*). If the thickness of ITZ is infinitely large, *t* → ∞, then the area of cement paste matrix tends to zero, *e_v_*(*t*) → 0. Therefore, the volume fraction of ITZ in corroded mortar can be expressed as follows:(19)$$f_{ITZ}=1-f_{s}-e_{v}\left(t\right)$$
where $f_{ITZ}$ and $f_{s}$ are the volume fraction of the ITZ and the sand, respectively.

When the thickness *t* of ITZ and the particle size distribution of sand are known, *e_v_*(*t*) can be obtained, as follows:(20)$$e_{v}\left(t\right)=(1-f_{s})\exp\left\{-2f_{s}S\left[a_{0}{\left(\frac{t}{\bar{D_{N}}}\right)}^{3}+a_{1}{\left(\frac{t}{\bar{D_{N}}}\right)}^{2}+a_{2}\left(\frac{t}{\bar{D_{N}}}\right)\right]\right\},\begin{array}{cc} & t\geq 0 \end{array}$$
(21)$$a_{0}=\left[4B\left(1-f_{s}\right)\left(1-f_{s}+3f_{s}S\right)+4Af_{s}{}^{2}S^{2}\right]/{\left(1-f_{s}\right)}^{3}$$
(22)$$a_{1}=\left[6B\left(1-f_{s}\right)+9f_{s}S\right]/{\left(1-f_{s}\right)}^{2}$$
(23)$$a_{2}=3/\left(1-f_{s}\right)$$
(24)$$B={\left(\bar{D_{N}}\right)}^{2}/\bar{D_{N}^{2}}$$
(25)$$S=\left(\bar{D_{N}^{2}}\times \bar{D_{N}}\right)/\bar{D_{N}^{3}}$$
where *t* is the thickness of ITZ in corroded mortar, *A* is a constant value (when the sand aggregate is assumed as circular, *A* = 0), $\bar{D_{N}}$ is the average diameter of fine aggregates, $\bar{D_{N}^{2}}$ is the second order origin moment of average diameter of fine aggregates, and $\bar{D_{N}^{3}}$ is the third-order origin moment of average diameter of fine aggregates. Equation (26) is as follows:(26)$$f_{c}=1-f_{s}-f_{ITZ}$$

Based on Lu and Torquato’s model and the mix proportion of mortar, the volume fractions of phases in the corroded mortar can be calculated as follows: (1) Based on the mix ratio of the mortar and the densities of sand ($\rho_{s}$ = 2.6 g/cm^3^), cement ($\rho_{c}$ = 3.10 g/cm^3^), and water ($\rho_{w}$ = 1 g/cm^3^), the volume fraction of sand aggregate was calculated by Equation (18). (2) Substituting the particle size distribution of sand and the thickness of the ITZ (20 µm) into Equation (19), the volume fraction of ITZ was obtained by Lu and Torquato’s model. (3) Finally, the volume fraction of cement paste matrix (not including ITZ) was calculated by Equation (26). The theoretical calculation results of volume fractions the of phases in corroded mortar are shown in Table 4.

## 5. Verification of Elastic Modulus Prediction Model of Mortar under Sulfate Corrosion

### 5.1. Theoretical Calculation of Macroscopic Elastic Modulus of Corroded Mortar

Based on the mechanical parameters of the phases in Section 4.2 and the volume fractions of the phases in Section 4.3, the theoretical calculation of the macro-elastic modulus of corroded mortar were obtained by the two-scale homogenization method in Section 4.1. The prediction results of the elastic modulus of corroded mortar are listed in Table 5 and Table 6, respectively.

### 5.2. Comparison between Experimental Results and Theoretical Prediction Results of the Macro-Elastic Modulus of Corroded Mortar

The error value between the theoretical and experimental values of the macro-elastic modulus of corroded mortar can be calculated by the following Equation (27):(27)$$\delta =\frac{\left|\bar{E_{t}}-E_{c}\right|}{E_{c}}\times 100\%$$
where δ is the error value between the theoretical value and the experimental value of the elastic modulus, $\bar{E_{t}}$ is the average value of experimental elastic modulus, and *E_c_* is the theoretical elastic modulus. The error values of mortar (M1 and M2) were 7.7% and 1.9%, respectively, and the error value of the elastic modulus predicted by the theoretical model is less than 8% compared with the actual elastic modulus obtained by the test. The data show that the theoretical calculation model proposed in this paper can well predict the elastic modulus of corroded mortar.

## 6. Conclusions

In this paper, the elastic parameters and the volume fractions of the phases (sand, cement, and ITZ) in mortar were obtained by tests and theoretical calculation. Then, a computational prediction model of the elastic modulus was established with homogenization methods to predict the elastic modulus of mortar under external sulfate attack. The conclusions are shown as follows:
(1)The BSE images of corroded mortar and the algorithm of image threshold segmenting can be used to acquire the thickness of ITZ, and the thickness of ITZ in corroded mortar at 88 d is 20 µm.(2)Combined with the results of nanoindentation test and the characteristics of corroded mortar, the macro-elastic modulus of ITZ in the corroded mortars can be accurately and effectively evaluated. The values of ITZ in corroded mortars (M1 and M2) are 15.2 GPa and 18.9 GPa, respectively.(3)Based on the mortar mix proportion and the thickness of ITZ, the volume fractions of sand, cement paste, and ITZ in corroded mortar can be well evaluated by Lu and Torquato’s model.(4)The computational elastic modulus prediction model, considering the influence of ITZ, can well predict the elastic modulus of mortar under sulfate attack.


## Figures and Tables

**Figure 1 materials-16-06167-f001:**
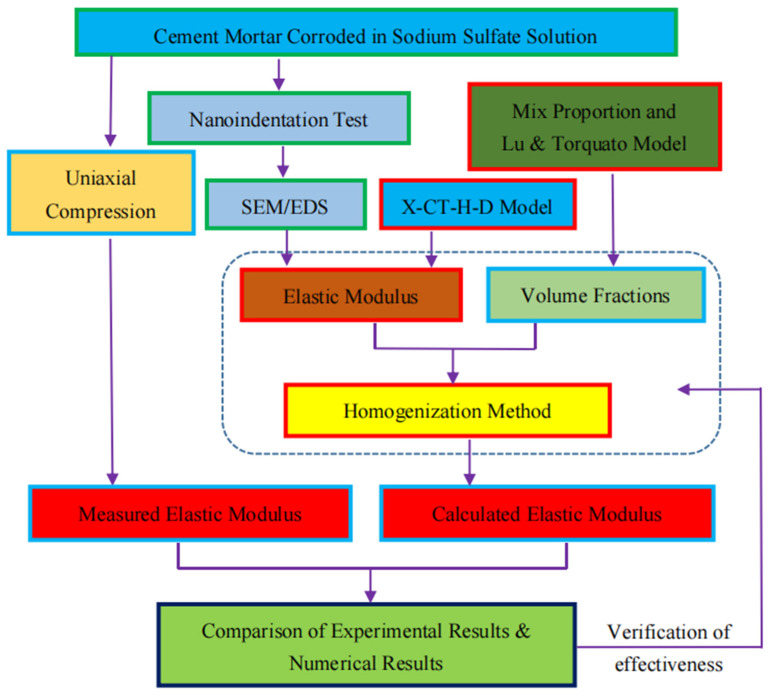
Roadmap of this paper.

**Figure 2 materials-16-06167-f002:**
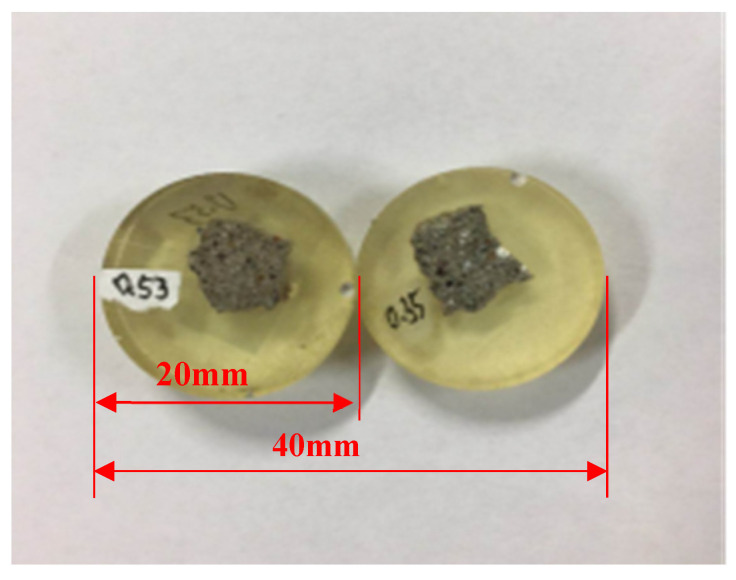
Specimens for nanoindentation test.

**Figure 3 materials-16-06167-f003:**
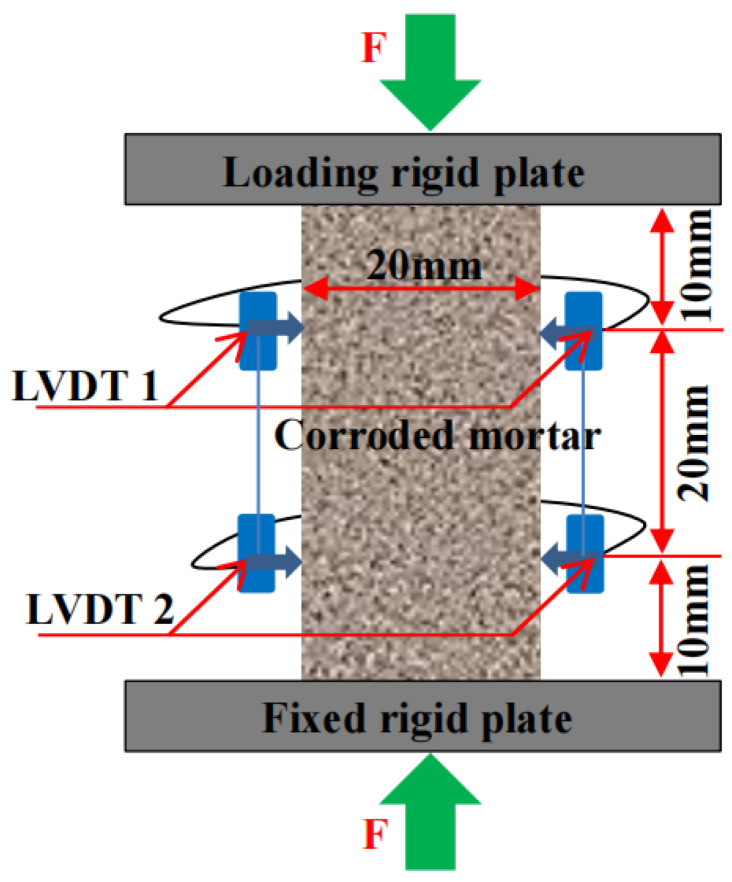
Uniaxial compression test of the mortar specimen.

**Figure 4 materials-16-06167-f004:**
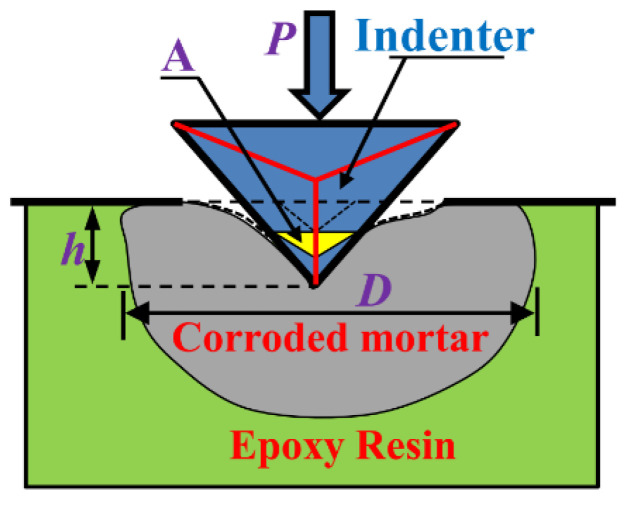
Schematic of nanoindentation test for the corroded mortar.

**Figure 5 materials-16-06167-f005:**
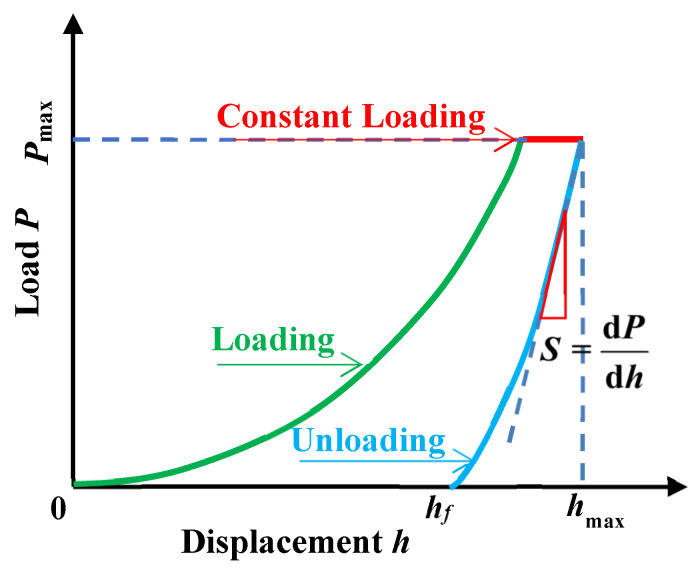
The load–displacement curve of a typical indentation point.

**Figure 6 materials-16-06167-f006:**
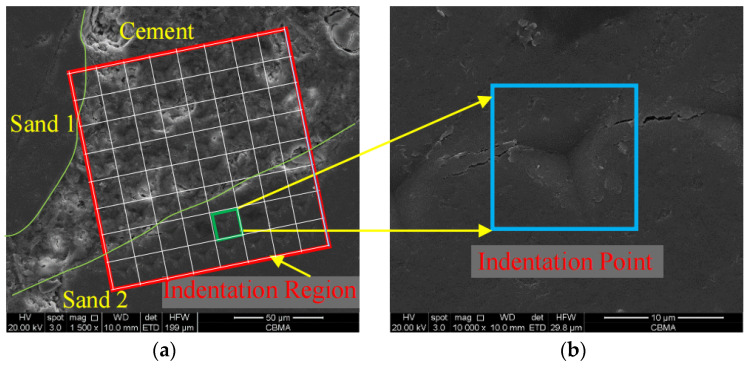
SEM-BSE of nanoindentation points. (**a**) Lattice map of nanoindentation points; (**b**) single nanoindentation point.

**Figure 7 materials-16-06167-f007:**
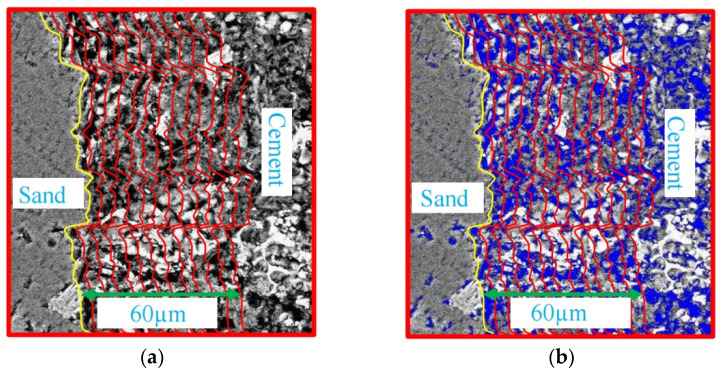
Image processing of ITZ. (**a**) SEM-BSE image; (**b**) image after threshold segmentation.

**Figure 8 materials-16-06167-f008:**
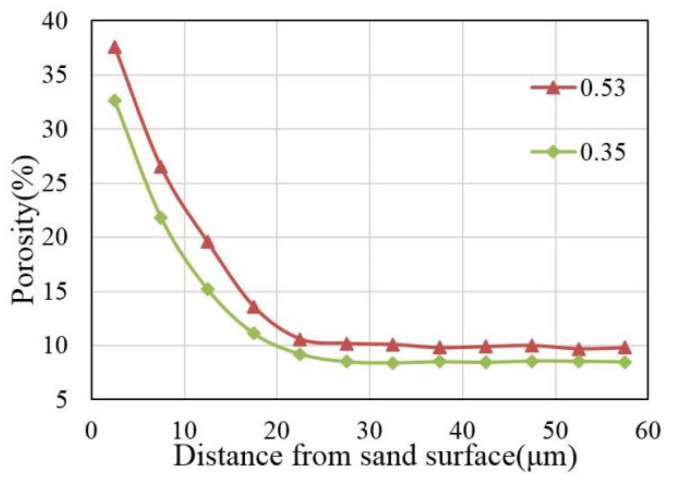
Variation in porosity with the increase in the distance from the sand edge.

**Figure 9 materials-16-06167-f009:**
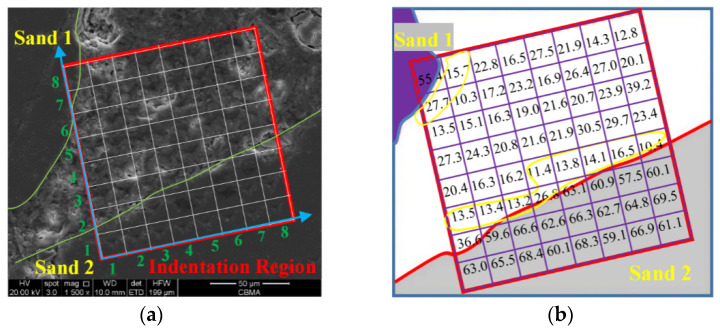
The schematic diagram of the nanoindentation results of corroded mortar (M1). (**a**) SEM-BSE image of the nanoindentation region; (**b**) microscopic elastic modulus of 8 × 8 nanoindentation points.

**Figure 10 materials-16-06167-f010:**
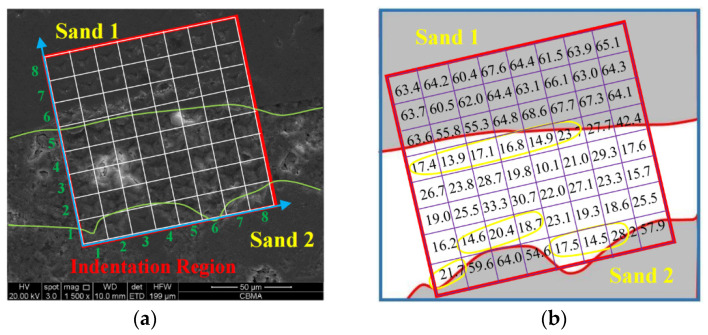
The schematic diagram of the nanoindentation results of corroded mortar (M2). (**a**) SEM image of nano indentation region; (**b**) microscopic elastic modulus of 8 × 8 nanoindentation points.

**Figure 11 materials-16-06167-f011:**
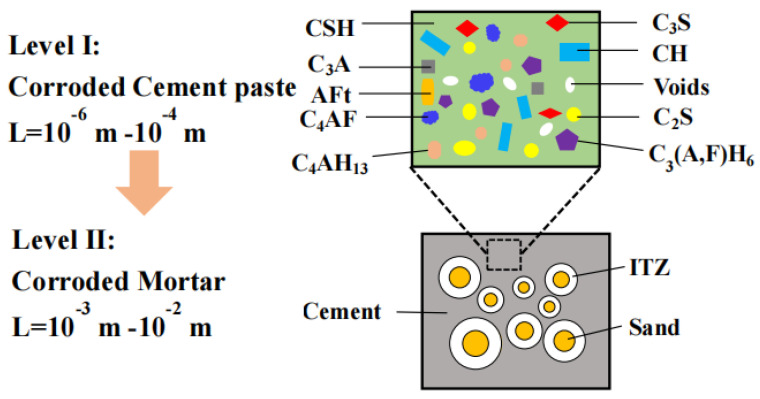
Multiscale microstructure of corroded mortar.

**Figure 12 materials-16-06167-f012:**
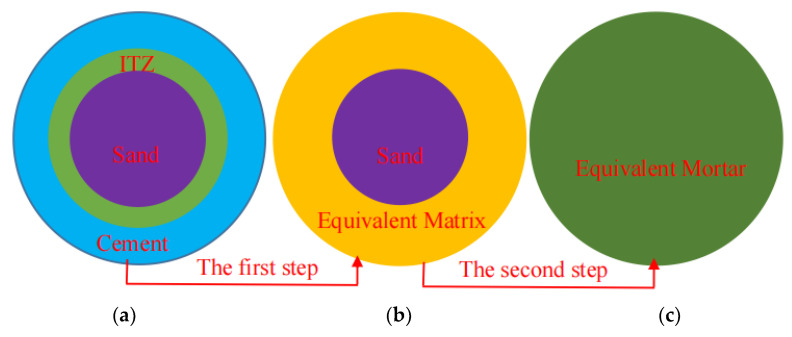
Two-step homogenization method. (**a**) Mortar; (**b**) equivalent result of the first step; (**c**) equivalent result of the second step.

**Figure 13 materials-16-06167-f013:**
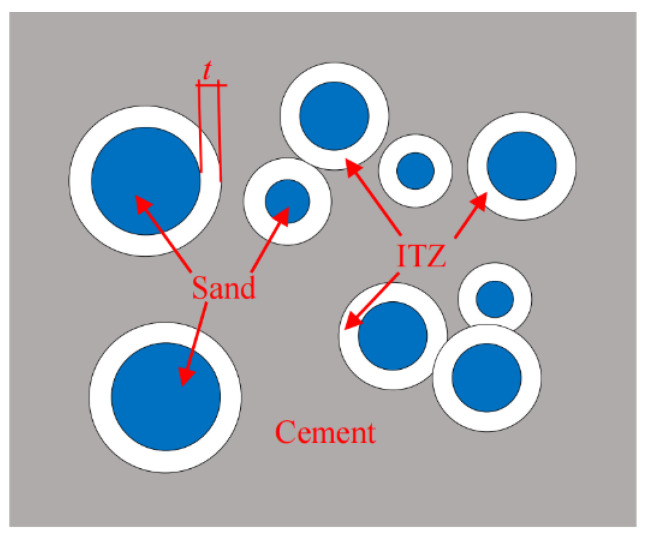
Schematic of Lu and Torquato’s model.

**Table 1 materials-16-06167-t001:** Mineral composition of cement.

Composition	C_3_A	C_2_S	C_3_S	C_4_AF
Content (%)	7.41	16.79	63.94	11.86

**Table 2 materials-16-06167-t002:** Mortar mixing ratio design.

Type	Water (g)	Cement (g)	Sand (g)	w/c
M1	400	750	1500	0.53
M2	312.2	900	1350	0.35

**Table 3 materials-16-06167-t003:** Theoretical calculation results of the elastic parameters of cement paste at 88 d (Gpa).

w/c	Bulk Modulus	Shear Modulus	Elastic Modulus
0.53	14.03	8.63	21.48
0.35	15.13	9.33	23.22

**Table 4 materials-16-06167-t004:** The theoretical calculation results of volume fractions of phases in mortar (%).

Type	Sand	Cement Paste	ITZ
M1	47.0	41.9	11.1
M2	46.0	43.1	10.9

**Table 5 materials-16-06167-t005:** Prediction results of elastic modulus of corroded mortar (M1).

Homogenization Process	Input Parameters			Output Results		
	Elastic Parameters		Volume Fractions	Elastic Parameters		
	k (GPa)	μ (GPa)	C (%)	k (GPa)	μ (GPa)	*E* (GPa)
Level I: Equivalent matrix (SC method)						
ITZ	9.74	6.13	20.94	12.98	8.04	19.99
Cement	14.03	8.63	79.06			
Level II: mortar (two-phase ball model)						
Equivalent matrix	12.98	8.04	53	17.90	13.51	32.38
Sand	29.28	27.74	47			

**Table 6 materials-16-06167-t006:** Prediction results of the elastic modulus of corroded mortar (M2).

Homogenization Process	Output Results			Output Results		
	Elastic Parameters		Volume Fractions	Elastic Parameters		
	k (GPa)	μ (GPa)	C (%)	k (GPa)	μ (GPa)	*E* (GPa)
Level I: Equivalent matrix (SC method)						
ITZ	12.12	7.62	20.55	14.45	8.95	22.25
Cement	15.13	9.33	79.45			
Level II: mortar (two -phase ball model)						
Equivalent matrix	14.45	8.95	54	18.84	14.16	33.97
Sand	29.28	27.74	47			

## Data Availability

Not applicable.

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
