# Peer review of "Experimental–Computational Investigation of the Elastic Modulus of Mortar under Sulfate Attack"

_materials, 2023, doi:10.3390/ma16186167_

Round 1

Reviewer 1 Report

I enjoyed reading this article, especially its very engineering and practical content, including identification methods. The language and style are fine. The research problem itself is clear, although one would expect some emphasis on new aspects in the last paragraph of the introduction. 

In my opinion, the review of the literature is extensive and up to date. However, in my opinion, the introduction lacks a more general view of different identification approaches. One can find many interesting identification techniques applied in many engineering fields in e.g.: researchgate.net/publication/278681276 or researchgate.net/publication/226145121 by Giulio Maier. I am sure that the Authors will find both suggestions interesting.

Major remark:

1) The word "experimental-computational" used in the title does not reflect the actual picture of the manuscript, at least to some extent. The title suggests that the problem has been solved by experimental methods, computer methods, and inverse methods (often called computational methods).
Perhaps a better solution would be to use in the title the formulation: experimental-analytical investigation?
I encourage the authors to get familiar with the methods described in the suggested works by prof. Maier, where one can find also the problem of indentation and nano-indentation.

2) In addition, the use of homogenization methods seems to be very simplified. Homogenization is usually based on mapping the behavior of the material, omitting the modeling of geometrical details (see e.g. works by Aleksander Marek on the homogenization of sandwich panels, or others also using inverse methods in homogenization).
In this manuscript, the authors used the analytical method of homogenization (developed by other authors). It is recommended, therefore, to introduce a broader view of homogenization methods and embed the method used here as one of the possibilities.

3) My last concern is the lack of emphasis on the novelty introduced by the authors in this work. If all the methods used by the authors are just adaptations of existing methods, what are the authors' contributions?

Minor

1) Figure 1 poor quality and font size should be slightly larger

2) Table 2 - what is Cenemt?

3) there is a missing space below Table 2

4) Equations 6-14 - almost all brackets are not scaled

Reviewer 2 Report

The reviewed article entitled “Experimental-Computational Investigation of Elastic Modulus of mortar under sulfate attack” shows a significant contribution related to the elastic modulus of mortar in extreme conditions. Some revisions should be carried out as follows:

1 - AbstractThe abstract should be improved and state objectives, methodology, and findings. The novelty of the article is not clear.

2 - The introduction needs to be improved. The importance of the subject is not evident. Why mortar? Why sulfate attack? Why elastic modulus?...

3 - The quality of the figures is very bad, for example in Figure 1 the text cannot be read. They should be improved.

4 - Table 1: How was determined the mineral composition of cement?

5 - Table 2: Please, include the units in the table.

6 - The authors should check typing errors throughout the manuscript. For example table 2 "cenemt".

7 - The conclusion should be more carefully rewritten, summarizing what has been learned and why it is interesting and useful.

Round 2

Reviewer 2 Report

The authors answered all of my questions and the article can now be accepted.

Reviewer 3 Report

I am glad that the authors effectively addressed my concerns. Therefore, based on the authors' satisfactory response, I find this version of the article to be acceptable.